# Myokine Musclin Is Critical for Exercise-Induced Cardiac Conditioning

**DOI:** 10.3390/ijms24076525

**Published:** 2023-03-30

**Authors:** Matthew P. Harris, Shemin Zeng, Zhiyong Zhu, Vitor A. Lira, Liping Yu, Denice M. Hodgson-Zingman, Leonid V. Zingman

**Affiliations:** 1Department of Internal Medicine, Fraternal Order of Eagles Diabetes Center, Abboud Cardiovascular Research Center, University of Iowa, Iowa City, IA 52242, USA; 2Veterans Affairs Medical Center, Iowa City, IA 52246, USA; 3Department of Health and Human Physiology, Fraternal Order of Eagles Diabetes Center, Pappajohn Biomedical Institute, University of Iowa, Iowa City, IA 52242, USA; 4NMR Core Facility and Department of Biochemistry, University of Iowa, Iowa City, IA 52242, USA

**Keywords:** preconditioning, ischemia, reperfusion, stress resistance, musclin, osteocrin, cardioprotection, exercise, heart disease, natriuretic peptide, pharmacokinetics

## Abstract

This study investigates the role and mechanisms by which the myokine musclin promotes exercise-induced cardiac conditioning. Exercise is one of the most powerful triggers of cardiac conditioning with proven benefits for healthy and diseased hearts. There is an emerging understanding that muscles produce and secrete myokines, which mediate local and systemic “crosstalk” to promote exercise tolerance and overall health, including cardiac conditioning. The myokine musclin, highly conserved across animal species, has been shown to be upregulated in response to physical activity. However, musclin effects on exercise-induced cardiac conditioning are not established. Following completion of a treadmill exercise protocol, wild type (WT) mice and mice with disruption of the musclin-encoding gene, *Ostn*, had their hearts extracted and exposed to an ex vivo ischemia-reperfusion protocol or biochemical studies. Disruption of musclin signaling abolished the ability of exercise to mitigate cardiac ischemic injury. This impaired cardioprotection was associated with reduced mitochondrial content and function linked to blunted cyclic guanosine monophosphate (cGMP) signaling. Genetic deletion of musclin reduced the nuclear abundance of protein kinase G (PKGI) and cyclic adenosine monophosphate (cAMP) response element binding (CREB), resulting in suppression of the master regulator of mitochondrial biogenesis, peroxisome proliferator-activated receptor γ coactivator 1α (PGC1α), and its downstream targets in response to physical activity. Synthetic musclin peptide pharmacokinetic parameters were defined and used to calculate the infusion rate necessary to maintain its plasma level comparable to that observed after exercise. This infusion was found to reproduce the cardioprotective benefits of exercise in sedentary WT and *Ostn*-KO mice. Musclin is essential for exercise-induced cardiac protection. Boosting musclin signaling might serve as a novel therapeutic strategy for cardioprotection.

## 1. Introduction

Significant advances in prevention and mitigation of cardiovascular disease have been made, but related morbidity and mortality remain high [1,2]. These adverse outcomes stem from two main processes: dysrhythmias and derangement of cardiac mechanical function, both related to ongoing cardiac structural and functional deterioration that is independent from the initial insult causing myocardial injury [3,4]. Cardiac conditioning is a therapeutic approach to induce endogenous protective adaptive responses in the heart that counteract myocardial loss or dysfunction [5]. Physical activity is one of the most powerful triggers of cardiac conditioning with proven benefits for both those with healthy and with diseased hearts [6,7,8,9,10,11,12,13,14,15,16,17,18,19]. The full breadth and complexity of cardioprotective mechanisms induced by exercise are still under investigation. A comprehensive exploration of the elements underlying exercise conditioning may reveal mechanisms that could be clinically harnessed to complement traditional cardiac risk prevention and treatment strategies, particularly when exercise activity is not possible or well-tolerated.

Myokines are cytokines or peptides produced and secreted by muscles. There is growing recognition that myokines mediate the local and systemic communication of muscles that is necessary to promote exercise tolerance and overall health, including cardiac conditioning [20,21]. We have previously demonstrated that the myokine musclin, highly conserved across animal species, is upregulated in response to physical activity and augments exercise endurance [22]. Further, musclin has been found to have a myocardial protective effect against injury from ischemia [23] and pressure overload [24]. Yet, despite these important observations and the vital physiological role implied by the evolutionary conservation of musclin expression across animal species [22,25,26,27], a musclin effect on exercise-induced cardiac conditioning has not previously been established.

The musclin peptide was discovered and described by two groups: one as a chondrocyte-derived osteocrin [28] and the other as a skeletal muscle-secreted musclin [29]. In mature mice, musclin production by skeletal muscle predominates [22] and is upregulated by physical activity [22]. Musclin contains a region homologous to members of the natriuretic peptide (NP) family [28,29] and has high affinity for the NPR_C_ clearance receptor, comparable to that of the cardiac NPs but, unlike ANP and BNP, very low affinity for guanylyl cyclase-coupled NPR_A_ and NPR_B_ [28,29,30,31]. Consistent with these biochemical characteristics [28,29,30,31], particularly musclin’s ability to limit clearance of NPs by competing with them for binding to NPR_C_ [30], we’ve shown that musclin, when combined with ANP, boosts cGMP signaling [22,23].

Here, to assess the role of the exercise-induced myokine musclin in promoting exercise-induced cardiac stress resistance, mice genetically engineered to lack musclin expression were subjected to treadmill exercise. This reveals that the absence of musclin signaling significantly attenuates the normal exercise-induced degree of cardiac conditioning and protection from myocardial injury in response to ischemia-reperfusion, as evidenced by larger myocardial infarcts in exercised musclin-deficient mice than exercised control mice. On the molecular level, disruption of normal musclin function results in significantly diminished exercise-triggered stimulation of cGMP/PKGI/CREB/PGC1α and reduced mitochondrial biogenesis in the myocardium. On the other hand, synthetic musclin infusion mimics the cardioprotective effects observed with exercise and can rescue the cardioprotective deficits in mice genetically lacking endogenous musclin production. Thus, this work identifies musclin as a critical mediator of the skeletal muscle-to-heart crosstalk that governs exercise-induced cardioprotection.

## 2. Results

### 2.1. Musclin Is Required for Exercise-Mediated Protection from Ischemia-Reperfusion Injury

It is well-established that exercise training significantly improves myocardial tolerance to ischemia-reperfusion (IR) injury [32,33,34,35,36,37,38]. Musclin is established as an exercise-induced myokine with cardioprotective properties [22,23], however its role in exercise-induced cardiac protection is unknown. Here, the effect of moderate intensity treadmill exercise training on myocardial resistance to IR injury was compared in WT control vs. a mouse model with ubiquitous disruption of the musclin-encoding gene, *Ostn* (*Ostn*-KO, genOway) [22]. Specifically, we used forced treadmill exercise, which combines elements of aerobic and resistance training [22] and was previously linked to enhanced musclin production and secretion [22]. As expected, exposure to exercise training of WT controls resulted in augmented myocardial stress resistance as evidenced by smaller infarct size in response to the IR protocol (Figure 1; 49.94 ± 8.761%, *n* = 11) compared to that of sedentary WT mice (59.12 ± 6.033%, *n* = 10, *p* = 0.031). Importantly, in *Ostn*-KO mice, exercise training failed to improve resistance to IR injury (60.39 ± 7.58%, *n* = 12) compared to sedentary *Ostn*-KO mice (59.67 ± 9.351%, *n* = 7, *p* = 0.99). Exercise-trained *Ostn*-KO exhibited significantly larger infarct sizes than exercise-trained WT mice (*p* = 0.0169; two-way ANOVA values for all comparisons between groups). Thus, intact musclin production is critical for exercise-related cardioprotection from ischemic injury.

### 2.2. Musclin Promotes Exercise Training-Induced Enhancements in Mitochondrial Biogenesis and Respiratory Capacity

It has been previously demonstrated that exercise-induced cardiac stress resistance is at least in part mediated through mitochondrial adaptation [39,40,41]. Musclin has been shown to govern exercise-induced mitochondrial biogenesis in skeletal muscles [22]. However, its effect on physical activity-triggered mitochondrial biogenesis in the heart is yet to be established. To this end, the hearts of sedentary or exercise-trained WT and *Ostn*-KO mice were assessed for mitochondrial content and oxidative phosphorylation potential.

First, protein expression of the subunits of the electron transport chain was measured in both sedentary and exercise-trained *Ostn*-KO and WT mice. In response to exercise training in WT mice there was a significant increase in complex II expression (0.88 ± 0.04 vs. 0.60 ± 0.12 AU, *n* = 4, *p* = 0.005), complex III expression (0.41 ± 0.13 vs. 0.13 ± 0.01 AU, *n* = 4, *p* = 0.005), complex IV expression (0.40 ± 0.05 vs. 0.21 ± 0.02 AU, *n* = 4, *p* = 0.0006), and complex V expression (0.85 ± 0.21 vs. 0.29 ± 0.07, *n* = 4, *p* = 0.002) complex I expression increased, but the change was not statistically significant (0.51 ± 0.18 vs. 0.31 ± 0.02, *n* = 4, p=0.07; Figure 2A,B). However, in *Ostn*-KO mice there was no exercise-induced upregulation of OXPHOS protein expression, and significantly lower expression of complex III in comparison to sedentary *Ostn*-KO mice (0.13 ± 0.01 vs. 0.32 ± 0.06 AU *n* = 4, *p* = 0.0002; Figure 2C,D). When comparing WT mice to *Ostn*-KO mice, hearts from exercise-trained WT mice, vs. exercise-trained *Ostn*-KO, had significantly higher expression of complex III (0.98 ± 0.12 vs. 0.83 ± 0.03 AU, *n* = 4, *p* = 0.0499) and complex IV subunits (0.88 ± 0.12 vs. 0.28 ± 0.03 AU, *n* = 4, *p* = 0.0001; Figure 2E,F).

Additionally, there was significantly lower mitochondrial protein content when normalized to wet heart weight in both sedentary and exercise-trained *Ostn*-KO mice compared to WT mice (5.85 ± 0.66 vs. 6.93 ± 0.52, *n* = 6 and *p* = 0.02; 6.67 ± 0.60 vs. 8.52 ± 1.08, *n* = 6 and *p* = 0.0003, respectively, in Figure 3A). Importantly, exercise training vs. sedentariness led to a significant increase in mitochondrial protein content only in WT (6.93 ± 0.52 vs. 8.52 ± 1.08, *n* = 6 and *p* = 0.0015), but not in *Ostn*-KO (5.85 ± 0.66 vs. 6.67 ± 0.60, *n* = 6 and *p* = 0.122; Figure 3A).

Next, mitochondrial DNA levels were quantified. Specifically, expression was measured for mitochondrial DNA encoded NADH dehydrogenase 1 (ND1) as a marker of mitochondrial biogenesis, normalized to hexokinase 2 as a marker of nuclear DNA. This showed that exercise training promotes increased ND1 expression in WT but not in *Ostn*-KO mice (2.46 ± 0.99 vs. 1.00 ± 0.40, *n* = 6 and *p* = 0.002; 1.08 ± 0.38 vs. 0.65 ± 0.09 *n* = 6 and *p* = 0.56, respectively, in Figure 3B). This physical activity-induced ND1 upregulation resulted in a significantly higher ND1 level in exercised WT mice compared to exercised *Ostn*-KO mice (1.08 ± 0.38 vs. 2.46 ± 0.99, *n* = 6 and *p* = 0.0003 in Figure 3B), while there was no difference in ND1 level between the two groups under sedentary conditions (0.65 ± 0.09 vs. 1.00 ± 0.40, *n* = 6 and *p* = 0.70 in Figure 3B).

To further ascertain the effect of musclin on exercise-induced cardiac mitochondrial oxidative capacity, citrate synthase activity was measured in isolated mitochondria and maximal mitochondrial respiration in semi-permeabilized cardiac fibers. While both WT and *Ostn*-KO mice significantly increased their citrate synthase activity with exercise training in mitochondrial extracts (13.59 ± 1.24 vs. 9.59 ± 1.01 AU, *n* = 6 and *p* < 0.0001; 10.12 ± 0.65 vs. 8.26 ± 0.97 AU, *n* = 6 and *p* = 0.046, respectively), citrate synthase activity was still significantly lower in exercise-trained *Ostn*-KO mice in comparison to exercise trained WT mice (10.12 ± 0.65 vs. 13.59 ± 1.24 AU, *n* = 6 and *p* = 0.004 in Figure 3C).

Similarly, in semi-permeabilized myofibers isolated from hearts, only those of WT mice but not those of *Ostn*-KO mice exhibited an increase in mitochondrial respiration in response to exercise (Figure 3D). Under carbohydrate-supported conditions, respiration in fibers from hearts of exercise-trained mice was significantly upregulated when stimulated with ADP alone (247.29 ± 75.27 vs. 125.56 ± 46.32 pmol*s^−1^/mg, *n* = 5 and *p* = 0.0006) or with the addition of succinate (279.68 ± 81.51 vs. 162.63 ± 33.89 pmol*s^−1^/mg, *n* = 5 and *p* = 0.001 in Figure 3D). In contrast, fibers from the hearts of exercise-trained *Ostn*-KO mice did not exhibit upregulation of mitochondrial respiration compared to sedentary controls with ADP stimulation alone (149.11 ± 53.25 vs. 137.88 ± 39.05 pmol*s^−1^/mg, *n* = 5 and *p* = 0.98) or with the addition of succinate (176.76 ± 55.81 vs. 177.17 ± 31.87 pmol*s^−1^/mg, *n* = 5 and *p* > 0.9999). Moreover, when compared to WT mice, *Ostn*-KO mice had significantly lower oxygen consumption when stimulated with ADP alone (149.11 ± 53.25 vs. 247.29 ± 75.27 pmol*s^−1^/mg, *n* = 5 and *p* = 0.007) or with the addition of succinate (176.76 ± 55.81 vs. 279.68 ± 81.51 pmol*s^−1^/mg, *n* = 5 and *p* = 0.004 in Figure 3D). Differences between sedentary WT and *Ostn*-KO mice were less marked (Appendix A), consistent with the finding that baseline (non-exercise) levels of musclin signaling are low in WT mice [22]. Thus, musclin signaling is essential for exercise-induced cardiac mitochondrial adaptations.

### 2.3. Musclin Regulates Cardiac cGMP/Protein Kinase G-Dependent Signaling

NP/cGMP signaling is recognized as a key regulator of mitochondrial biogenesis, and, given that we previously demonstrated the role of musclin in augmenting ANP/cGMP signaling in skeletal muscle [22], and others described this in the heart [23,24,42], we hypothesized that musclin would influence exercise-induced cardiac cGMP signaling. To assess this hypothesis, cGMP levels were first measured in the myocardium of exercised WT and *Ostn*-KO mice. We confirmed that exercise-trained WT mice had significantly higher cardiac cGMP following 8 days of treadmill exercise compared to *Ostn*-KO mice (98.51 ± 14.16 vs. 61.94 ± 19.66 pMol/mg, *n* = 5 and *p* = 0.01). cGMP is known to activate protein kinase G (PKG) [43], which can translocate into the nucleus (and phosphorylates cAMP response element binding (CREB) in baby hamster kidney (BHK) fibroblasts [44,45,46] promoting its nuclear localization and ability to regulate transcription of peroxisome proliferator-activated receptor γ coactivator 1α (PGC1α) [47,48]. To investigate if musclin affects this signaling cascade in the heart, the nuclear expressions of PKG1, CREB1, and PGC1α were quantified. In WT hearts collected immediately after last exercise session compared to those of sedentary WT controls, a significant increase was observed in nuclear localization of PKG1 (1.43 ± 0.13 vs. 0.76 ± 0.12 AU, *n* = 4 and *p* = 0.0003), CREB1 (1.87 ± 0.44 vs. 0.83 ± 0.01 AU, *n* = 4 and *p* = 0.31), and PGC1α (0.57 ± 0.11 vs. 0.28 ± 0.01 AU, *n* = 4, *p* = 0.002, Figure 4A,B). However, in *Ostn*-KO mice, exercise did not exhibit the same degree of upregulation of PKG1 (2.24 ± 1.07 vs. 1.59 ± 0.84 AU, *n* = 4 and *p* = 0.37), CREB1 (3.11 ± 0.64 vs. 2.46 ± 1.38 AU, *n* = 4 and *p* = 0.43), or PGC1α (0.76 ± 0.20 vs. 0.73 ± 0.17 AU, *n* = 4 and *p* = 0.85 in Figure 4C,D). Additionally, when comparing exercised groups, *Ostn*-KO had significantly lower nuclear expression than WT of PKG1 (0.21 ± 0.01 vs. 0.29 ± 0.01 AU, *n* = 4 and *p* < 0.0001), CREB1 (0.69 ± 0.10 vs. 0.89 ± 0.05 AU, *n* = 5 and *p* = 0.01), and PGC1α (1.16 ± 0.19 vs. 1.63 ± 0.32 AU, *n* = 4 and *p* = 0.04 in Figure 4E,F).

Similarly, in response to exercise in WT mice, compared to sedentary WT controls, there was upregulation in cardiac mRNA expression of *Creb1* (1.73 ± 0.25 vs. 1.00 ± 0.06 fold change, *n* = 5 and *p* = 0.0002) and *PPARG Coactivator 1 Alpha* (*Ppargc1a*) (1.69 ± 0.09 vs. 1.01 ± 0.16 fold change, *n* = 5 and *p* < 0.00004) as well as their downstream targets linked to mitochondrial biogenesis: *Estrogen Related Receptor Alpha* (*Erra*) (1.52 ± 0.15 vs. 1.05 ± 0.37 fold change, *n* = 5 and *p* = 0.029), *Nuclear Respiratory Factor 2* (*Nrf2*) (1.50 ± 0.22 vs. 1.02 ± 0.21 fold change, *n* = 5 and *p* = 0.007), *and Transcription Factor A, Mitochondrial* (*Tfam*) (1.56 ± 0.31 vs. 1.01 ± 0.19 fold change, *n* = 5 and *p* = 0.011) but not *Nuclear Respiratory Factor 1* (*Nrf1*) (1.21 ± 0.18 vs. 1.01 ± 0.14, *n* = 5 and *p* = 0.08 in Figure 5A). This upregulation in gene expression was not observed in response to exercise training in *Ostn*-KO mice: *Creb1* (1.26 ± 0.13 vs. 1.02 ± 0.25 fold change, *n* = 5 and *p* = 0.096), *Ppargc1a* (1.34 ± 0.25 vs. 1.03 ± 0.27 fold change, *n* = 5, 0.1), *Erra* (1.27 ± 0.22 vs. 1.02 ± 0.22 fold change, *n* = 5 and *p* = 0.11), *Nrf1* (1.32 ± 0.25 vs. 1.02 ± 0.21 fold change, *n* = 5 and *p* = 0.07), *Nrf2* (1.49 ± 0.16 vs. 1.04 ± 0.29 fold change, *n* = 5 and *p* = 0.02), and *Tfam* (1.22 ± 0.13 vs. 1.02 ± 0.23 fold change, *n* = 5 and *p* = 0.13 in Figure 5B). Additionally, in exercised WT mice compared to *Ostn*-KO mice, there was significantly higher expression of *Creb1* (1.59 ± 0.34 vs. 1.00 ± 0.10 fold change, *n* = 5, *p* = 0.007), *Ppargc1a* (1.40 ± 0.08 vs. 1.01 ± 0.19 fold change, *n* = 5 and *p* = 0.003), *Erra* (1.30 ± 0.08 vs. 1.01 ± 0.18 fold change, *n* = 5 and *p* = 0.009), *Nrf2* (1.31 ± 0.22 vs. 1.01 ± 0.11 fold change, *n* = 5 and *p* = 0.03), and *Tfam* (1.41 ± 0.22 vs. 1.01 ± 0.11 fold change, *n* = 5 and *p* = 0.006), but not *Nrf1* (1.27 ± 0.16 vs. 1.09 ± 0.21 fold change, *n* = 5 and *p* = 0.17 in Figure 5C). Hearts of sedentary *Ostn*-KO and WT mice exhibited similar levels of nuclear PKG1 expression (0.69 ± 0.12 vs. 0.88 ± 0.22 AU, *n* = 4 and *p* < 0.18 in Appendix A) and mRNA levels of *Creb1* (0.93 ± 0.22 vs. 1.00 ± 0.06 fold change, *n* = 5 and *p* = 0.49), *Ppargc1a* (0.96 ± 0.27 vs. 1.01 ± 0.16 fold change, *n* = 5 and *p* = 0.77), *Erra* (0.97 ± 0.21 vs. 1.05 ± 0.37 fold change, *n* = 5 and *p* = 0.68), *Nrf1* (0.99 ± 0.19 vs. 0.95 ± 0.19 fold change, *n* = 5 and *p* = 0.78), and *Tfam* (0.94 ± 0.21 vs. 1.01 ± 0.20 fold change, *n* = 5 and *p* = 0.60 in Appendix A), Only expression of *Nrf2* was slightly lower in sedentary *Ostn*-KO mice (0.82 ± 0.17 vs. 1.09 ± 0.15 fold change, *n* = 5 and *p* = 0.03 in Appendix A). Taken together, these results indicate that musclin enhances physical activity-induced cardiac mitochondrial biogenesis through augmentation of cGMP/PKG/CREB-dependent signaling.

### 2.4. Synthetic Musclin Infusion Mimics Exercise-Mediated Protection against IR Injury

Exercise is one of the most effective ways to improve cardiac function and stress resistance [6,7,8,9,10,11,12,13,15,16,17,18]. However, adherence among those with heart disease to even short exercise programs, such as cardiac rehabilitation, is suboptimal [49]. The reasons are multifactorial, but a major contributor is baseline exercise intolerance, which is common in those with heart disease [7,50]. Synthetic musclin infusion has been demonstrated to be effective for reduction of myocardial injury and recovery of cardiac function when used after coronary artery ligation [23], but the possibility of using musclin infusion as a therapeutic strategy to induce a profile of benefits overlapping with exercise-induced cardiac conditioning in order to prevent myocardial damage and loss of cardiomyocytes is an intriguing possibility. To test this concept, synthetic peptide, corresponding to aa 80–112 of the murine musclin [22,30], was tested. This fragment has been associated with the most prominent modulation of ANP signaling in previous studies [22,30].

First, this synthetic peptide, containing uniformly labeled with ^13^C and ^15^N alanine and valine at positions 26 and 27, was used to define the volume of distribution (V_d_) and plasma elimination half-life based on the calculations described in the Methods section (Figure 6A–E). These parameters were used to calculate the infusion rate necessary to maintain a musclin plasma level comparable with endogenously produced levels observed after exercise training [22] (Figure 6A–E).

Musclin treatment for 14 days significantly reduced myocardial infarct size following IR in sedentary WT mice compared to treatment with the control peptide (48.39 ± 2.15 vs. 59.82 ± 1.91%, *n* = 7–8 and *p* = 0.002 in Figure 7A,C). While 14 days of musclin infusion did not significantly reduce myocardial infarct size in sedentary *Ostn*-KO mice (Figure 7B,C), 28 days of treatment reduced infarct size following IR (48.57 ± 2.14 vs. 59.69 ± 3.30%, *n* = 7–8 and *p* = 0.013 in Figure 7B,C). Importantly, enhanced cardiac tolerance to IR injury after musclin infusion was associated with markers of increased mitochondrial biogenesis (Appendix A). These results demonstrate that exogenous musclin mimics exercise-induced cardiac conditioning and stress resistance.

## 3. Discussion

This study establishes that the exercise-responsive myokine musclin [22] is vital for exercise-induced cardiac resistance to IR stress. Disruption of musclin signaling in *Ostn*-KO mice results in reduced oxidative phosphorylation potential and greater vulnerability to ischemic injury. Musclin replacement therapy restored cardiac resistance to stress in *Ostn*-KO mice. These findings indicate a previously unrecognized pathway for cardiac adaptation to exercise.

Several studies indicate that exercise training improves myocardial tolerance to IR injury (e.g., myocardial infarction in both young and old, male and female, animals) [32,33,34,36,37,51,52,53,54,55,56,57,58,59,60,61,62,63,64,65,66,67,68,69]. Specifically, studies reveal that exercise training protects the heart against arrhythmias, oxidative injury, mitochondrial damage, and cell death [33,55]. Interestingly, several investigators have shown that short-term exercise training (3–5 consecutive days) provides the same cardioprotection as that observed following long-term (10 weeks) training [34,51,52,55,56,57,62,63,70,71]. Here, we use forced treadmill exercise which combines elements of aerobic and resistance training [22]. We previously demonstrated that this exercise protocol is associated with enhanced musclin production and secretion [22], which in turn modulates atrial natriuretic peptide (ANP)/cyclic GMP signaling [22] in skeletal muscles. The current study confirms that intact musclin function promotes myocardial cGMP signaling triggered by exercise in the heart.

The data in the present study also confirm that intact musclin signaling is necessary for exercise-induced activation of PKGI-CREB/PGC1α, mitochondrial biogenesis, and functional adaptations in the myocardium. Notably, the results indicate significantly greater mitochondrial quantity in WT compared to musclin-deficient hearts by multiple methods and demonstrate the functional importance of the observed changes by revealing a significant deficiency in respiration measured in perforated myofibers isolated from *Ostn*-KO mice. These findings agree with the previously reported musclin interaction with natriuretic peptide receptors [23,24,30,31,42] and the detection of significantly higher levels of cGMP in the myocardium of WT mice, compared with *Ostn*-KO mice, after an exercise bout. There are accumulating data suggesting cGMP is a critical driver of mitochondrial biogenesis [72,73,74,75,76,77,78]. In fact, cGMP signaling has been linked to PGC1α–dependent mitochondrial biogenesis in many studies in different tissues and organs, including our own prior study indicating musclin-dependent activation of cGMP/PGC1α-driven mitochondrial biogenesis in skeletal muscles [22], while the current data indicate the significance of this musclin-driven signaling in the heart (Figure 8).

Importantly, different time points after the last exercise training session were used here to reveal specific features of exercise-induced cardiac conditioning. Hearts were collected immediately after exercise for cGMP measurement and 10–15 min after exercise to evaluate the nuclear presence of PKGI and CREB, while nuclear expression of PGC1α and mitochondrial respiratory complexes were evaluated in hearts collected 2 h after the last exercise session to probe the previously reported dynamics of cGMP/PKGI signaling [44,79,80].

Although treadmill exercise training was used here to stimulate musclin signaling, results indicate a phenotype of lower mitochondrial content and reduced mitochondrial respiration even in untrained *Ostn*-KO mice compared with WT controls. Importantly, these observed trends and changes in sedentary *Ostn*-KO became much more obvious and significant after exposure to the exercise protocol. These findings can be interpreted to indicate that musclin production and secretion, although more easily demonstrated in response to vigorous exercise, are physiologically relevant, even for routine daily activities.

Cardioprotective properties of musclin were demonstrated previously in different murine models of cardiomyopathies [10,23,81,82]. Our work demonstrates the role of musclin in cardiac conditioning and protection that is specifically exercise-induced. This establishes a previously unrecognized link between muscle activity and cardiac conditioning and supports a therapeutic potential in critically ill, injured, or otherwise high cardiac-risk surgical candidates in whom significant loss of muscle mass due to the catabolic stress of immobility and/or inadequate nutrition associated with any critical illness [83] may undermine optimal intrinsic musclin signaling.

In fact, the data from this study indicate that infusion of a synthetic fragment of musclin peptide mimics exercise benefits with respect to protection of the myocardium from IR injury. This is very encouraging as pharmacological options available to mimic exercise-induced cardioprotection have not previously been clinically available. In contrast to ANP or BNP infusion, which can directly stimulate cGMP synthesis, musclin promotes cGMP signaling through inhibition of other NPs via competitive inhibition of NPR_C_-driven clearance [22,23]. In this way, musclin infusion would enhance the natural dynamics of NP signaling as well as any potential beneficial effects related to local signaling by multiple NPs that are cleared through the NPR_C_ [84], in contrast to systemic infusion of ANP and BNP that would presumably result in non-physiologically high levels in off-targets and possibly insufficient levels in target tissues such as the myocardium. Interestingly, inhibition of another pathway for NP degradation by neutral endopeptidase has been linked to a survival benefit in heart failure (HF) patients [85]. In contrast, ANP or BNP infusion, while leading to improvement in measures of HF, has been linked to an increased risk of side-effects related to hypotension or nephrotoxicity [86,87,88], supporting the concept that off-target effects of direct infusion of natriuretic peptides are less desirable than methods to boost their intrinsic local concentrations and effects.

The present study also includes pharmacokinetic studies that allow for the calculation of the infusion rate necessary to secure a plasma musclin concentration close to that previously linked to exercise training and enhanced cGMP signaling [22]. As such, the findings of this study may help standardize future experiments with synthetic musclin.

Surprisingly, disruption of musclin signaling in *Ostn*-KO mice practically abolished exercise-induced cardioprotection. Considering the complexity of exercise-induced adaptations, this is somewhat unexpected. However, this finding may occur since musclin is also important for the skeletal muscle response to physical activity and, as such, *Ostn*-KO mice may not be getting the full benefits of exercise training [22]. Furthermore, this study investigates the role of musclin in augmenting NP/cGMP signaling. However, there is emerging research that indicates the existence of NPR_c_—specific signaling [89,90,91]. This possible musclin/NPR_c_ interaction should be the subject of future studies.

Finally, this study further verifies the unique interaction of skeletal muscle and heart through the previously reported co-regulation of endocrine signals: cardiac NPs and the myokine musclin [22,42]. Cardiac NPs, particularly ANP, have been increasingly recognized as powerful exercise-responsive regulators of important physiological functions [92,93,94,95]. Uncovering this musclin-dependent effect on exercise-induced cardiac conditioning broadens appreciation of the complexity and physiologic roles of cardiac and skeletal muscles as endocrine organs. In the future it will be interesting to clarify the effect of musclin signaling on other NP-regulated physiologic phenomenon such as blood pressure homeostasis as well as glucose and fat metabolism [84,96,97]. Additional investigation, including direct comparison of different exercise modes (resistance vs. aerobic) as well as forced vs. voluntary exercise, along with different time exposures, are needed to fully understand the range of musclin-induced adaptations that underlie its beneficial cardioprotective effects. Beyond advancing the understanding of musclin’s effects and musclin’s role in mechanisms of exercise-induced cardiac conditioning, this study has translational implications by extending the concept of musclin-dependent cardioprotection to the preconditioning of hearts to improve their tolerance of future ischemic insults using synthetic musclin infusion. This could represent a new therapeutic strategy to improve outcomes of stressful cardiac events or procedures, especially in exercise-intolerant patients.

## 4. Methods

### 4.1. Animal Experiments

All animal protocols conform to the Guide for the Care and Use of Laboratory Animals generated by the Institute for Laboratory Animal Research, National Research Council of the National Academies. All animal protocols were approved by the University of Iowa Institutional Animal Care and Use Committee. For all experiments, mice were anesthetized with inhaled isoflurane (5% induction, 1–1.5% maintenance; Piramal Healthcare) to maintain a respiratory rate of ~50–60 breaths per minute.

### 4.2. Ostn-KO Mouse Model

*Ostn*-KO mice were generated as described previously [22]. Vector construction and targeted knockout strategy were designed together with genOway: Mice were generated based on deletion of a 2.1-kb sequence flanking *Ostn* exon 2, resulting in inactivation of the ATG and signal peptide. All mice used in this study were male, 6–8 weeks of age, and on a C57BL6/N background. Control WT C57BL6/N were originally purchased from Charles River (Wilmington, MA, USA). Heterozygous *Ostn*-KO mice were bred to obtain homozygous *Ostn*-KO mice and WT controls.

### 4.3. Myocardial Ischemia–Reperfusion Injury

Hearts were extracted from anesthetized mice and retrogradely perfused at 90 mm Hg with Krebs–Henseleit buffer bubbled with 95% O_2_/5% CO_2_, at 37 °C and pH 7.4. Myocardial ischemia-reperfusion (IR) injury was created as follows: after allowing 20 min for stabilization, the hearts were subjected to 20 min of global ischemia and 45 min of reperfusion. At the end of this treatment, the hearts were removed from the Langendorff perfusion apparatus and immediately frozen at −20 °C. The frozen hearts were then cut into transverse slices of approximately equal thickness (∼0.8 mm) from apex to base. The slices were placed into a small cell culture dish and then incubated in 1% triphenyltetrazolium chloride (TTC) in phosphate buffer (Na_2_HPO_4_ 88 mmol/L, NaH_2_PO_4_ 1.8 mmol/L, pH 7.8) at 37 °C for 20 min while gently shaking the dish. The development of the red formazan pigment in living tissues relies on the presence of lactate dehydrogenase, or NADH, while failure to stain red indicates a loss of these constituents from necrotic tissue. After staining, the TTC buffer was replaced by 10% formaldehyde. The slices were fixed for the next 4–6 h, and then the areas of infarct tissue were quantified using ImageJ software. The risk area is taken as the total ventricular cross-sectional area of all slices. The infarct size is presented as infarcted percentage of at-risk area.

### 4.4. Treadmill Exercise Training

Three days before exercise, mice were acclimated for 45 min daily on a non-moving treadmill (Columbus Instruments, Columbus, OH, USA) followed by 15 min at a velocity of 3.5 m/min at 15° incline. For IR studies and assessment of nuclear PKG and PGC1α, mice were then exercised daily for total of 10 days: five consecutive days (Monday to Friday), followed by a 2-day rest, and then exercised daily for three more days (Monday, Tuesday, Wednesday) at a speed of 15 m/min and inclination of 15° for 45 min. This protocol has been previously shown to increase the production and secretion of musclin [22]. Immediately after the last exercise session, mice were sacrificed, and their hearts were collected for cGMP measurements. To evaluate the nuclear presence of PKGI and CREB, hearts were collected 10–15 min after the last exercise. To test nuclear expression of PGC1α and mitochondrial respiratory complexes, hearts were collected 2 h after the last exercise. Mitochondrial DNA, mRNA, protein expression, and mitochondrial respiration changes were assessed following 15 days of exercise training as described above (5 d/week daily exercise for 3 wks) with hearts collected 2 h after the last exercise.

### 4.5. Determination of Synthetic Musclin Peptide Plasma Elimination Half-Life and Volume of Distribution

The musclin [^13^C,^15^N-A26-V27]-labeled peptide sequence used was as follows: SFSGFGSPLDRLSAGSVEHRGKQRKAVDHSKKR, corresponding to amino acids 80–112; GenBank accession no. AAS87598.1 (Bio-synthesis, Inc. Lewisville, TX, USA). A peptide was synthesized with 98% purity, counter-ion exchange (TFA to Cl), and pH adjustment (Bio-synthesis, Inc. Lewisville, TX, USA).

For infusion, 1.7 mg of the chemically synthesized [^13^C,^15^N-A26-V27]-labeled musclin peptide was dissolved in 300 µL of PBS. This musclin solution was injected into the jugular vein. A 100 µL blood sample was then drawn from the carotid artery at 1, 3, 5, 7, 10, and 15 min after the peptide infusion. Blood was mixed with 5 µL of 10% NaEDTA and 1 µL of peptidase inhibitor cocktail (Sigma, St. Louis, MI, USA) and kept on ice, then centrifuged at 4 °C for 45 min at 14,000 rpm. Plasma was removed and stored at −80 °C for NMR studies.

### 4.6. NMR Spectroscopy

All NMR spectra were acquired at 5 °C on a Bruker Avance II 800 MHz NMR spectrometer equipped with a TCI cryoprobe. 100 µL of plasma (see above) was added 150 µL NMR buffer containing 50 mM sodium phosphate, pH 7.1 with a final volume of 250 µL in 8.5% D_2_O/91.5% H_2_O. This sample was then transferred to a Shigemi NMR tube for NMR studies. One-dimensional (1D) ^1^H, two-dimensional (2D) ^15^N/^1^H-heteronuclear single-quantum correlation (HSQC), 2D ^13^C/^1^H-HSQC with constant time experiment (HSQC-CT), and 2D ^15^N/^1^H-slice of the three-dimensional (3D) HNCO experiment [98,99] were acquired for each blood sample drawn at different times after peptide infusion. The collected NMR data were processed using NMRPipe [100] and analyzed using NMRView [101]. The musclin peptide concentration present in these blood samples was calibrated by using a known amount of chemically synthesized [^13^C,^15^N-A26-V27]-labeled musclin peptide. The peak intensity of these acquired NMR spectra was measured and fitted to the equation for half-life in exponential decay using GraphPad Prism (GraphPad Software version 9.4.1).

### 4.7. Pharmacokinetic Calculations

Peptide MW: 3649.675Initial concentration: 36 µM = 0.131 mg/mLQ—infusion rateC_ss_—steady state concentrationCl—clearanceT_1/2_—half-lifeK_e_—elimination rate = 0.693/T_1/2_ = 0.151 min^−1^V_d_—volume of distribution = administered dose/initial concentration1.7 mg/0.131 mg/mL = 12.997 mL for mice of 31 g → 0.419 mL/g of BWDesired steady state concentration: 200 pg/mLCl = K_e_ × V_d_ = 0.151 min^−1^ × 0.419 mL/g × 27 g = 1.71 mL/minQ = C_ss_ × Cl = 200 pg/mL × 1.71 mL/min = 341.652 pg/min → 20.28 ng/hFor mice of 27 g:Q = 174.636 pg/min → 10.478 ng/h → 3520.608 ng for 2 weeksFor osmotic pumps with 0.25 µL/h → 20.28 × 4 = 81.966 ng/µL → 8.197 µg per pump

### 4.8. Osmotic Pumps

Osmotic pumps (model 1002 or 1004, 100 µL volume, 0.25 or 0.11 µL/h release rate, 14–28 d duration; (Alzet Durect) were loaded with saline vs. mouse musclin vs. control peptide. Musclin peptide, corresponding to amino acids 80–112 of full length musclin; GenBank accession no. AAS87598.1. Control peptide has point mutations at amino acid residues that have homology with atrial natriuretic peptide (ANP) (Asp89Gly and Arg90Gly) [23]. Each amino acid sequence is as follows: musclin **SFSGFGSPLDRLSAGSVEHRGKQRKAVDHSKKR**, and control **SFSGFGSPLGGLSAGSVEHRGKQRKAVDHSKKR**.

Peptides dissolved in PBS were loaded into osmotic pumps, based on the pharmacokinetic data obtained with labeled peptide (see above). Pumps were surgically implanted within the peritoneal cavity of mice under sterile conditions and general anesthesia.

### 4.9. Mitochondrial Protein Isolation

Mitochondrial protein fraction was isolated from hearts by using the Mitochondria Isolation Kit (Thermo Scientific, Waltham, MA, USA) according to the manufacturer’s protocol. Mitochondrial protein content was determined as the amount of mitochondrial protein per mg of wet tissue.

### 4.10. RNA Isolation and qRT-PCR

Total RNA from cardiac tissues was isolated by using the RNeasy RNA Isolation Kit (Qiagen). A 1 µg primary myocyte or tissue RNA sample was used to synthesize cDNA in 50 µL reactions using Oligo(dT) as primer and SuperScript III Reverse Transcriptase (Invitrogen). Quantitative real-time PCR was performed on a Mastercycler EP Gradient S (Eppendorf) by using SYBR green-based PCRs. For qRT-PCR, 1 µL of reverse transcription reaction was mixed with 10 pmol of each specific primer and 12.5 µL of SYBR PCR Master Mix (Bio-Rad, Hercules, CA, USA). The reaction was incubated for 40 cycles consisting of denaturation at 95 °C for 10 s and annealing/extension at 59.9 °C for HPRT and Nrf2, and 56 °C for Creb1, ERRA, PGC1α, Tfam, and Nrf1 for 1 min. The quality of the PCR product was routinely checked by a thermal denaturation curve following the qPCR reactions. The threshold cycle (CT) was determined by Realplex2 software (Eppendorf), and quantification of relative mRNA levels was performed by ∆∆CT method or by calculation of the absolute number of copies based on the standard curve. The primers used in this study are as follows: HPRT forward (F), GGA CCT CTC GAA GTG TTG GAT AC, and HPRT reverse (R), GCT CAT CTT AGG CTT TGT ATT TGG CT; Creb1: Creb1F, CAG TAC ATT GCC ATT ACC CAG G, and Creb1R, GCA CTA GAA TCT GCT GTC CA; ERRA: ERRAF, GGA AGA CAG CCC CAG TGA, and ERRAR, AGT GAC AGT GAG GAG AAG CC; PGC1α: PGC1-αF, TGA TGT GAA TGA CTT GGA TAC AGA CA, and PGC1αR, GCT CAT TGT TGT ACT GGT TGG ATA TG; TfamF, GGA ATG TGG AGC GTG CTA AAA, and TfamR, TGC TGG AAA AAC ACT TCG GAA TA; Nrf1F, CGC AGC ACC TTT GGA GAA, and Nrf1R, CCC GAC CTG TGG AAT ACT TG; and Nrf2F, CAG CTC AAG GGC ACA GTG C, and Nrf2R,– GTG GCC CAA GTC TTG CTC C.

### 4.11. Western Blotting

Whole-cell protein extracts were prepared by homogenizing cardiac tissue in RIPA Buffer (Sigma), supplemented with protease and phosphatase inhibitors (Roche Inc. Basel, Switzerland). Nuclear extracts were obtained by using the Subcellular Protein Fractionation Kit for Tissues (Thermo Scientific). Electrophoresis was performed on 3–8% gradient Nu-Page Tris-Acetate or 10% Nu-Page Bis-Tris gels and transferred to 0.2 μm Sequi-Blot PVDF membranes (Bio-Rad). The membranes were blotted with total Creb1 (Cell Signaling, Danvers, MA, USA), PKG1 (Cell Signaling), total OXPHOS (Abcam, Cambridge, UK), PGC1α (Abcam), TBP, Lamin B1, and GAPDH (Cell Signaling) antibodies. Densitometric analysis of Western blots was performed using Adobe Photoshop (Adobe Systems, San Jose, CA, USA). For representative individual gel/blots, the reference proteins (laminB1 and TBP) are shown immediately below the proteins of interest.

### 4.12. CGMP Measurement

cGMP levels were measured using the cyclic GMP enzyme immunoassay (EIA) kit (Cayman Chemical, Ann Arbor, MI). On the day of the experiment, mice were exercised at 12 m/min for 20 min while sedated with isoflurane, and heart tissue was collected immediately at the end of exercise. Cyclic nucleotides were extracted by using 5% (wt/vol) trichloroacetic acid. Samples were acetylated and used for cGMP level measurements according to the manufacturer’s protocol.

### 4.13. Mitochondrial Respiration in Isolated Myofibers

Mitochondria respiratory function was assessed using an Oroboros Instruments Oxygraph 2K (Innsbruck, Austria) as described previously [102]. Cardiac fibers from a small piece of the endocardium (~15 mg) were dissected and separated into bundles in ice-cold BIOPS buffer (7.23 mM K2EGTA, 2.77 mM CaK2EGTA, 20 mM imidazole, Sigma-Aldrich, I2399), 0.5 mM DTT, 20 mM taurine, 5.7 mM ATP (Sigma-Aldrich, St. Louis, MI, USA, A7699), 14.3 mM phosphocreatine (Sigma-Aldrich, P7936), 6.56 mM MgCl_2_·6H_2_O, and 50 mM MES hydrate, pH 7.1). For permeabilization, fibers were then transferred to 1 mL of BIOPS buffer containing 40 µg/mL of saponin (Sigma-Aldrich, S7900) and gently mixed for 30 min at 4 °C. After this point, fibers were then transferred to 1 mL of respiration buffer (105 mM K-MES, 30 mM KCl, 10 mM K_2_HPO_4_, 5 mM MgCl_2_·6H_2_O, 1 mM EGTA, 20 µm blebbistatin (Selleck Chem, Houston, TX, USA; S7099), 2.5 mg/mL fatty acid free BSA (Sigma-Aldrich, A8806, pH 7.4), and gently mixed for an additional 10 min at 4 °C. Following incubation, 1–1.5 mg of tissue was added into the O2K chamber, which was filled with 2 mL of respiration buffer. Carbohydrate (CHO)-supported respiration was assessed with pyruvate (5 mM) and malate (2 mM) with subsequent addition of ADP (5 mM) and glutamate (5 mM). Maximal CHO-supported respiratory capacity was determined in the presence of these substrates with the subsequent addition of succinate (5 mM).

### 4.14. Mitochondrial DNA

DNA was isolated by using QIAamp DNA Mini Kit (no. 51034, Qiagen). Mitochondrial DNA (mtDNA) contents were determined by quantitating expression of a mitochondria-encoded gene, ND1 (forward: 5′-CTAGCAGAAACAAACCGGGC-3′; reverse: 5′-CCGGCTGCGTATTCTACGTT-3′) and a nuclear-encoded gene, hexokinase 2 (forward: 5′-GCCAGCCTCTCCTGATTTTAGTGT-3′; reverse: 5′-GGGAACACAAAAGACCTCTTCTGG-3′), as described previously [103]. The expression of ND1 was normalized to the expression of hexokinase 2.

### 4.15. Citrate Synthase Activity

Citrate synthase activity was determined using a colorimetric assay based on the reaction between 5,5′-dithiobis-(2-nitrobenzoic acid) (DTNB) and CoA-SH to form 5-thio-2-nitrobenzoic acid (TNB), which exhibits maximum absorbance at 412 nm, according to the manufacturer’s instructions (Sigma). A mitochondrial pellet fraction was isolated from the heart by using the Mitochondria Isolation Kit (Thermo Scientific) and resuspended in CS Assay Buffer.

### 4.16. Statistics

Data are presented as individual data points with means ± the standard deviation. Statistical significance was evaluated using two-way ANOVA with Tukey’s testing for multiple comparisons or a two-sided Student’s *t*-test with tests for normality/lognormality as appropriate (GraphPad Prism version 9.4).

## 5. Conclusions

The exercise-responsive myokine musclin, which has high homology to natriuretic peptides (NPs), is critical for exercise-induced myocardial protection.Normal musclin signaling is critical for exercise-induced cardiac mitochondrial biogenesis.Musclin-dependent cardioprotective remodeling is driven by activation of cGMP/PKGI/CREB/PGC1α signaling.Uncovering a musclin-dependent effect on exercise-induced cardiac conditioning broadens the appreciation of the complexity and physiologic roles of cardiac and skeletal muscles as endocrine organs.Synthetic musclin infusion reproduces the exercise-induced cardioprotective response to ischemia-reperfusion.

## Figures and Tables

**Figure 1 ijms-24-06525-f001:**
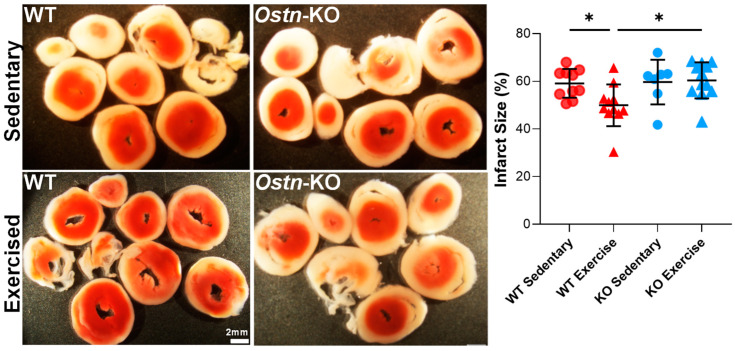
**Musclin is required for exercise-mediated protection from ischemia-reperfusion injury.** Representative transverse sections of individual TTC-stained hearts (**Left**) (scale bar 2 mm) and summary statistics of infarct size (**Right**) from sedentary and exercised WT and *Ostn*-KO mice (*n* = 7–12). Data are means ± SD; * *p* < 0.05.

**Figure 2 ijms-24-06525-f002:**
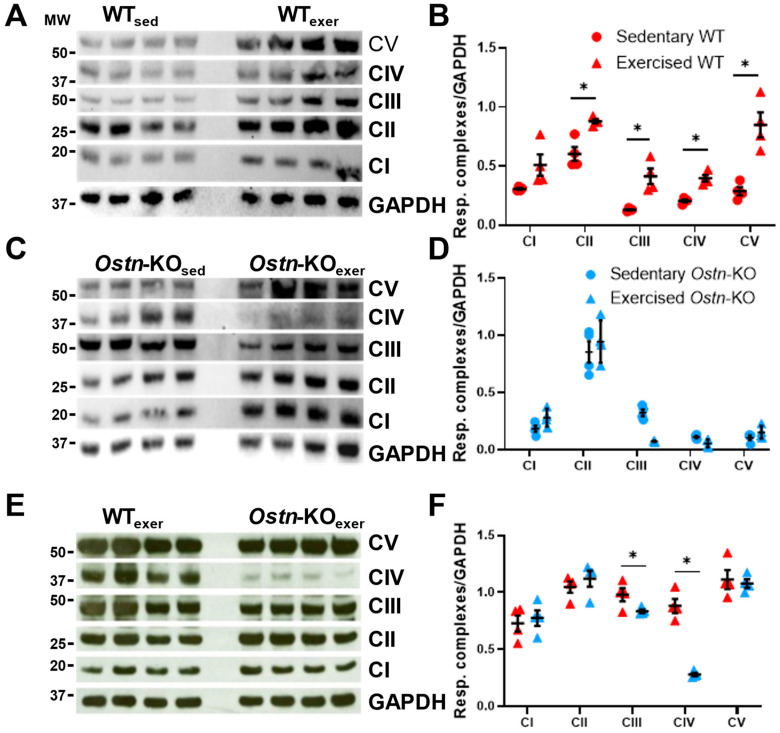
**Musclin promotes exercise training-induced enhancements in respiratory chain enzyme expression.** (**A**) Representative western blots of respiratory chain enzymes and GAPDH, and (**B**) summary statistics for respiratory complex expression normalized to GAPDH in the hearts of sedentary and exercised WT mice (*n* = 4). (**C**) Representative western blots of respiratory chain enzymes and GAPDH, and (**D**) summary statistics for respiratory complex expression normalized to GAPDH in the hearts of sedentary and exercised *Ostn*-KO mice (*n* = 4). (**E**) Representative western blots of respiratory chain enzymes and GAPDH, and (**F**) summary statistics for respiratory complex expression normalized to GAPDH in the hearts of exercised WT and *Ostn*-KO mice (*n* = 4). Data are means ± SD; * *p* < 0.05.

**Figure 3 ijms-24-06525-f003:**
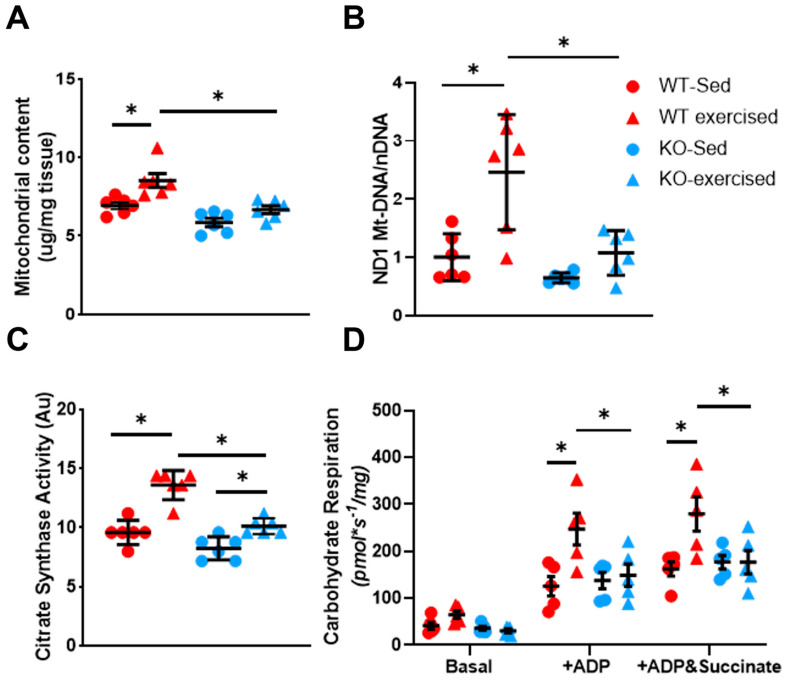
**Musclin promotes exercise training-induced enhancements in mitochondrial biogenesis and respiratory capacity.** (**A**) Summary statistics for mitochondrial content by tissue weight in hearts from sedentary and exercised WT and *Ostn*-KO mice (*n* = 6). (**B**) Summary statistics for expression of ND-1 normalized to hexokinase 2 in hearts from sedentary and exercise WT and *Ostn*-KO mice (*n* = 6). (**C**) Summary statistics for citrate synthase activity in hearts from sedentary and exercised WT and *Ostn*-KO mice (*n* = 6). (**D**) Summary statistics for carbohydrate-driven mitochondrial respiration in semi-permeabilized fibers normalized to tissue weight in hearts from sedentary and exercised WT and *Ostn*-KO mice (*n* = 5). Data are means ± SD; * *p* < 0.05.

**Figure 4 ijms-24-06525-f004:**
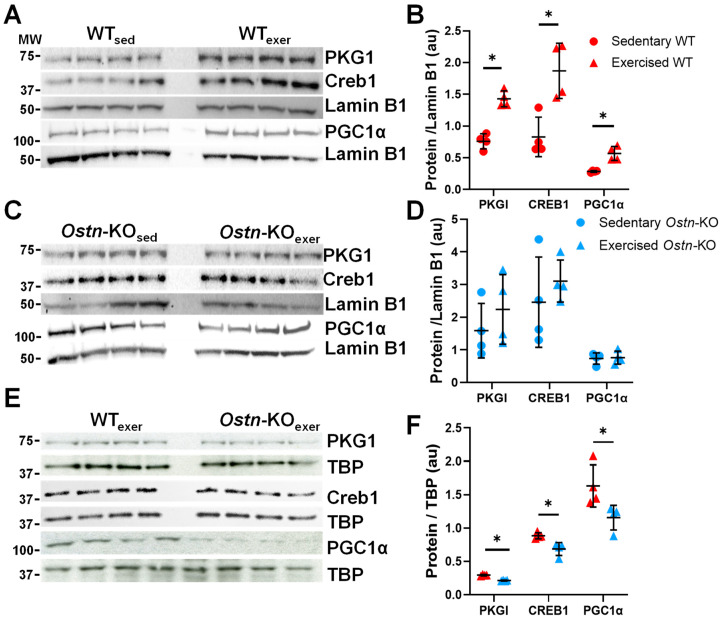
**Musclin regulates cardiac cGMP/Protein Kinase G-dependent signaling.** (**A**) Representative western blots of PKG1, CREB1, PGC1α and Lamin B1 and (**B**) summary statistics for PKG1, CREB1, and PGC1α expression normalized to Lamin B1 in nuclear extracts from hearts of sedentary and exercised WT mice (*n* = 4). (**C**) Representative western blots of PKG1, CREB1, PGC1α and Lamin B1 and (**D**) summary statistics for PKG1, CREB1, and PGC1α expression normalized to Lamin B1 in nuclear extracts from hearts of sedentary and exercised *Ostn*-KO mice (*n* = 4). (**E**) Representative western blots of PKG1, CREB1, PGC1α and TATA binding protein (TBP) and (**F**) summary statistics for PKG1, CREB1, and PGC1α expression normalized to TBP in nuclear extracts from hearts of exercised WT and *Ostn*-KO mice (*n* = 4). Data are means ± SD; * *p* < 0.05.

**Figure 5 ijms-24-06525-f005:**
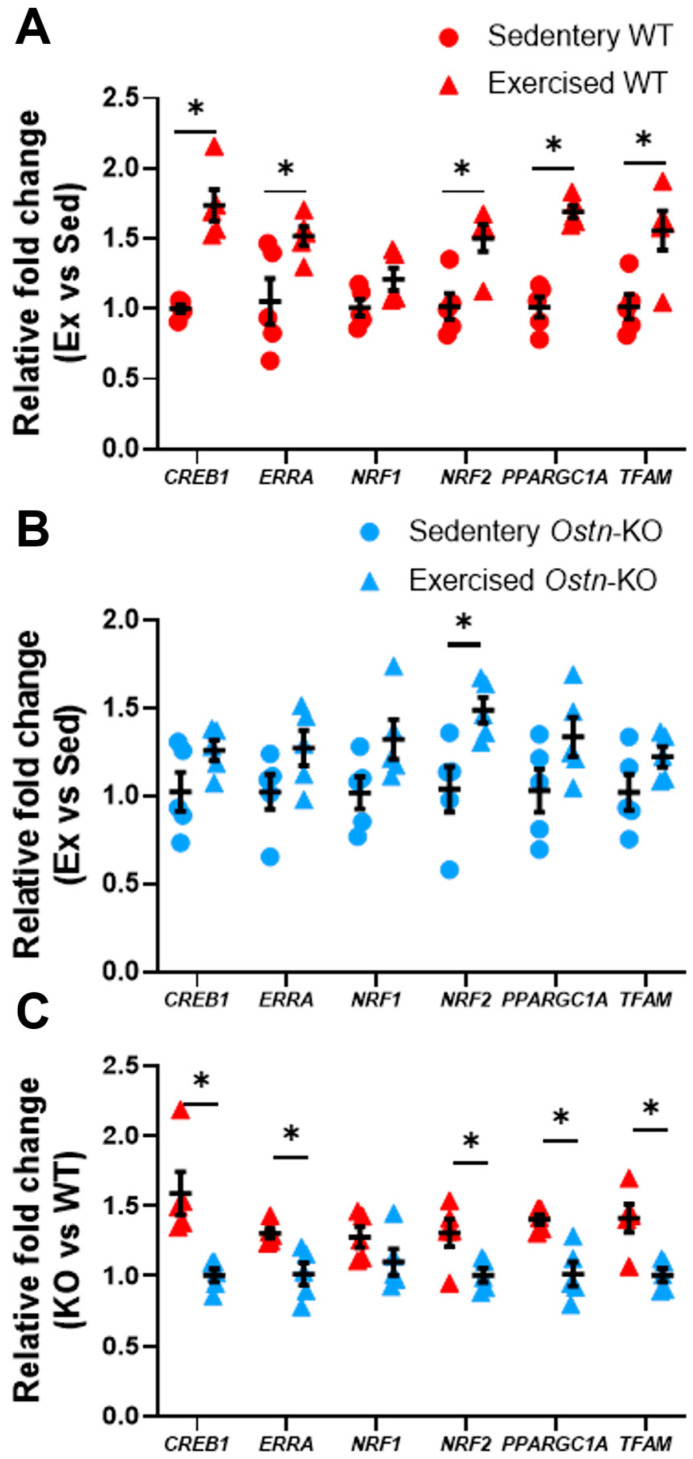
**Musclin upregulates cardiac transcription of genes involved in mitochondrial biogenesis.** Summary statistics for mRNA expression of *Creb1*, *Erra, Nrf1, Nrf2, Ppargc1a,* and *Tfam* in hearts from (**A**) sedentary and exercised WT mice (*n* = 6); (**B**) sedentary and exercised *Ostn*-KO mice (*n* = 6); and (**C**) exercised WT and *Ostn*-KO mice (*n* = 6). Data are means ± SD; * *p* < 0.05.

**Figure 6 ijms-24-06525-f006:**
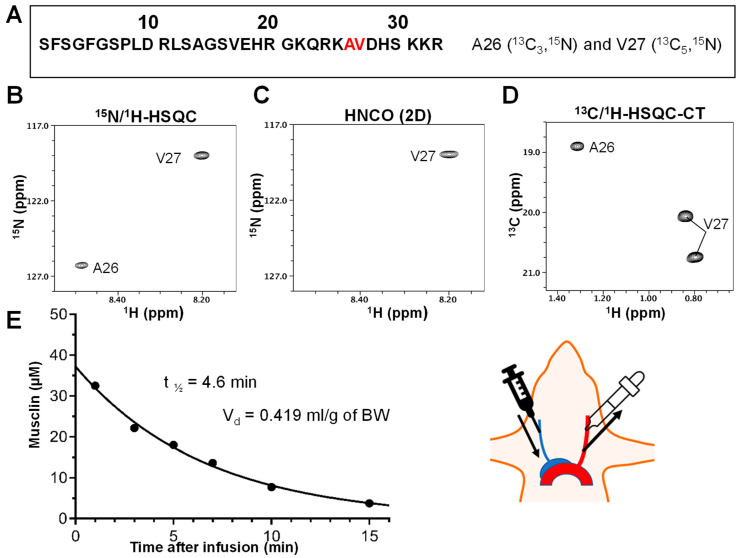
**Synthetic musclin infusion pharmacokinetics.** (**A**) Chemically synthesized [^13^C,^15^N-A26-V27]-labeled musclin peptide. (**B**) ^15^N/^1^H-HSQC spectrum. (**C**) 2D slice of the HNCO spectrum. (**D**) ^13^C/^1^H-HSQC-CT spectrum. (**E**) Half-life is determined from the data fit of the peak intensities measured in the HNCO (2D) spectra. Similar half-life values were also obtained from data in the ^15^N/^1^H-HSQC or ^13^C/^1^H-HSQC-CT spectra.

**Figure 7 ijms-24-06525-f007:**
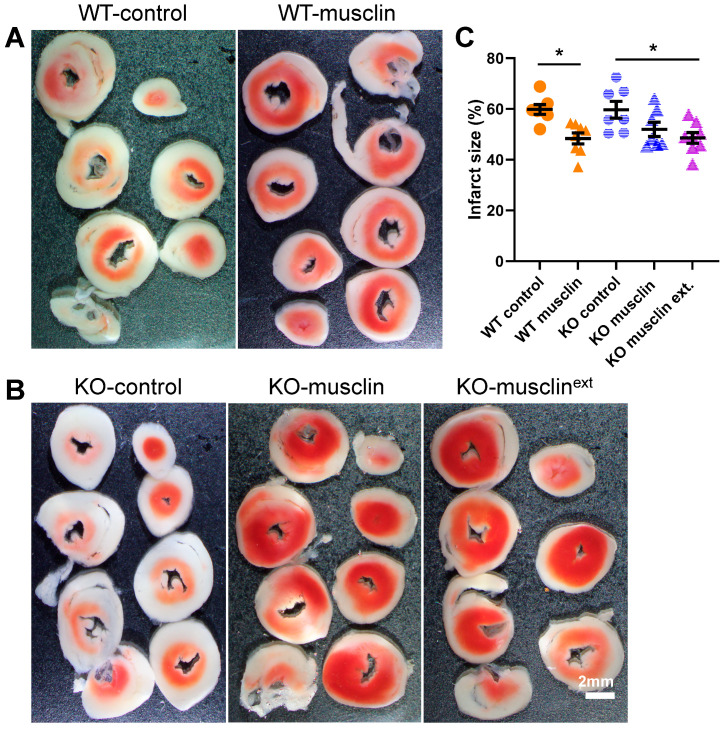
**Synthetic musclin infusion mimics exercise-mediated protection against IR injury**. (**A**,**B**)—Representative transverse sections of TTC stained hearts (scale bar = 2 mm) and (**C**) summary statistics of infarct size in WT and *Ostn*-KO mice treated with osmotic pumps containing control or musclin peptide for 14 days or 28 (musclin ext) days (*n* = 7–8). Data are means ± SD; * *p* < 0.05.

**Figure 8 ijms-24-06525-f008:**
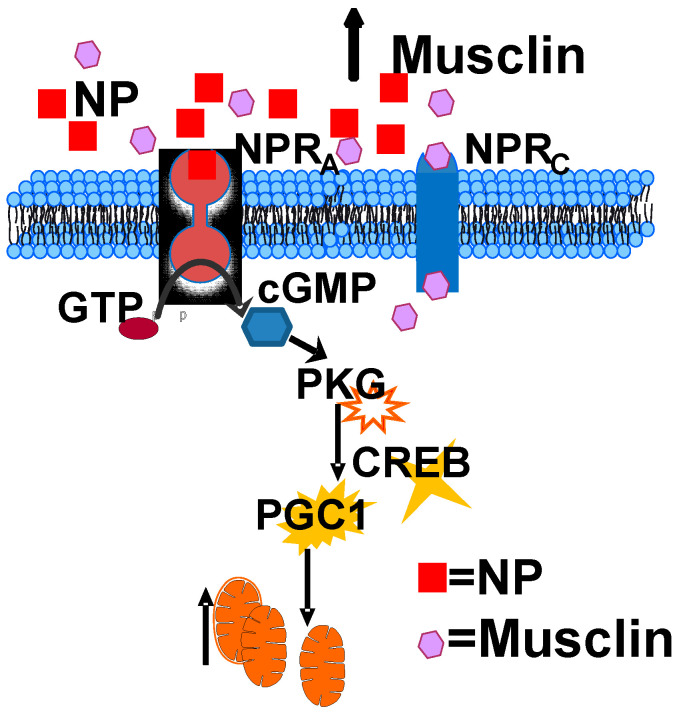
**Molecular mechanisms by which exercise-induced musclin promotes cardiac conditioning.** In the heart, musclin competitively interferes with binding to NPR_c_, allowing increased NP/NPR_a_ binding, which in turn increases cGMP production in the heart. cGMP in turn activates PKG1 which leads to a signaling cascade that results in increased CREB1 nuclear localization and transcription of PGC1α which promotes enhancement of mitochondrial biogenesis and respiratory capacity that support cardiac stress resistance.

## Data Availability

The data presented in this study are available in the article.

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
