# Peer review of "Myokine Musclin Is Critical for Exercise-Induced Cardiac Conditioning"

_ijms, 2023, doi:10.3390/ijms24076525_

Round 1

Reviewer 1 Report

Exercise training is one of the most effective interventions to reduce the severity of ischemic heart disease. In this study, Harris et al, aim to understand the mechanism underlying the cardioprotective effect of exercise.  For this, Ostn KO mouse lacking musclin is used to elucidate the role of musclin in promoting exercise-induced cardioprotection. Complementary approaches, including ex vivo IR injury, molecular biology, and biochemical experiments demonstrate that musclin is critical for activation of cGMP signaling, and mitochondrial adaptations in the cardiac tissue in response to exercise. The study also includes the rescue experiment with synthetic musclin infusions which reinstated the exercise effect and attenuated tissue damage in response to IR.

Overall, the paper offers a clear and compelling insight into the potential effectiveness of musclin on exercise-induced cardiac conditioning. The manuscript is well written and clear. The experimental parts exploring the musclin effects on exercise-induced cardiac conditioning are extensive, and the findings are consistent. However, the parts, assessing the effect of musclin on cGMP signaling following the IR injury appear to fall a bit short of providing conclusive evidence.

Points to address:

1   1. Effect of exercise on musclin expression is shown in the author’s previous study. Did authors checked the level of musclin in SED and exercise mice following IR injury. This western blot experiment should have also been performed with the Ostn knockout mice subjected to IR injury.

2   2. Figure 3: Representative respiration traces of cardiac fibers from SED and exercise WT and Ostn-KO mice should be provided. How much ADP was used in the mitochondrial respiratory measurement?

3    3. Figure 4: The authors have demonstrated that musclin regulates cardiac cGMP/PKG signaling at the basal/endogenous level. How was the mRNA and protein expression of PKG1, CREB1 and PGC1α following IR injury in SED and exercise WT and Ostn-KO mice?

2    4. What was cardiac cGMP level after IR injury in in SED and exercise WT and Ostn-KO mice?

Minor issue

5   5. Text from abstract (Line 13-17) is repeated in Introduction (Line 48-53).

Author Response

Authors’ response to reviewers: We thank the reviewers for their supportive comments, interesting questions, as well as critiques that helped us to significantly revise and improve the manuscript. This includes the addition of three supplemental figures as well as changes to the text to clarify the methods and results. The itemized responses to specific questions and critiques are as follows:

Reviewer #1:

1) Effect of exercise on musclin expression is shown in the author’s previous study. Did authors checked the level of musclin in SED and exercise mice following IR injury. This western blot experiment should have also been performed with the Ostn knockout mice subjected to IR injury.

Authors’ response: This study is focused on role of exercise-related musclin production/secretion in exercise-induced cardiac conditioning that precedes an ischemia/reperfusion stress, rather than the acute effects of IR on musclin production. We previously found and published that musclin is not expressed in the heart (PMID: 26668395). However, whether IR injury would acutely affect systemic or cardiac musclin levels due to altered skeletal muscle production and secretion of musclin, perhaps due to systemic effects of the ischemia such as hypotension or adrenergic stress that could affect skeletal muscle perfusion, is an interesting question. Since our IR experiment is done ex vivo in isolated, Langendorff perfused hearts, the acute effect of the IR stress model on skeletal muscle musclin expression/secretion cannot be performed. This would need to be tested in an in-vivo IR model and would be beyond the scope of the current study that is focused on the “chronic” pre-conditioning exercise-related musclin production/secretion effects rather than acute IR effects on skeletal musclin production/secretion. Further, an in-vivo model would not be able to distinguish between such chronic versus acute effects on the myocardium and thus would be less useful to address the central question of our study which relates to the pre-conditioning effects of exercise.

Also, unrelated to exercise, the effect of musclin signaling and overexpression on myocardial injury has been previously published by other groups (PMID: 35013221; 29326144).

      2) Figure 3: Representative respiration traces of cardiac fibers from SED and exercise WT and Ostn-KO mice should be provided. How much ADP was used in the mitochondrial respiratory measurement?

Authors’ response: Unfortunately, our instrument does not provide the ability to digitize the traces and, as such, the figure reflects the usual way this data is presentation as seen in other publications (PubMed PMID: 36442857; 36364742; 35104184).

We also thank the reviewer for noting the failure to include the concentration of ADP in the method section which has now been corrected (5mM).

      3) Figure 4: The authors have demonstrated that musclin regulates cardiac cGMP/PKG signaling at the basal/endogenous level. How was the mRNA and protein expression of PKG1, CREB1 and PGC1α following IR injury in SED and exercise WT and Ostn-KO mice?

What was cardiac cGMP level after IR injury in in SED and exercise WT and Ostn-KO mice?

Authors’ response: As in the response to the first critique, the IR injury experiments were done ex-vivo. Also, they were conducted on a time scale such that changes in myocardial mRNA expression would be minimal and dissociated from musclin signaling since any musclin that remained present locally upon excision of the heart would be expected to be washed out during the experiment.

      4) Text from abstract (Line 13-17) is repeated in Introduction (Line 48-53).

Authors’ response: We have now corrected this with the introduction now reading as follows: “There is growing recognition that myokines mediate the local and systemic communication of muscles that is necessary to promote exercise tolerance and overall health, including cardiac conditioning.”

Reviewer 2 Report

In this study, the authors investigated the role and mechanisms by which the myokine musclin promotes exercise-induced cardiac conditioning. They identified that disruption of musclin signaling abolished the ability of exercise to mitigate cardiac ischemic injury, and the impaired cardioprotection was associated with reduced mitochondrial protein, DNA, and functional capacity. Moreover, they found that genetic deletion of musclin reduced nuclear abundance of PKG, CREB, resulting in suppression of master regulator of mitochondrial biogenesis-PGC1α and its downstream targets. They concluded that musclin is essential for exercise-induced cardiac protection, and boosting musclin signaling might serve as a novel therapeutic strategy for cardioprotection in patients with impaired exercise tolerance. Overall, the current result is not solid enough to support the conclusion, and the molecular mechanism is not well studied.

I have the following concerns over the study, which requires a major revision.

1. Is there any difference between WTsed and Ostn-KOsed groups in figures 2, 4 and 5? The authors need to compare between these two groups.

2. Does synthetic musclin infusion has any effect on respiratory chain enzymes and respiratory capacity, mitochondrial biogenesis and cGMP/PKG signaling? The authors need to verify the molecular mechanism in vivo.

3. The phosphorylation of CREB is important for its function, does musclin has any effect on the phosphorylation of CREB?

4. The authors mentioned that musclin competitively interferes with binding to NRPc, allowing increased NP/NRPa binding, however, the experimental evidence is totally missing.

5. The expression level of LaminB1 in Figures 4A and 4C is not consistent.

6. The format of the references needs to be adjusted.

Author Response

Authors’ response to reviewers: We thank the reviewers for their supportive comments, interesting questions, as well as critiques that helped us to significantly revise and improve the manuscript. This includes the addition of three supplemental figures as well as changes to the text to clarify the methods and results. The itemized responses to specific questions and critiques are as follows:

1) Is there any difference between WTsed and Ostn-KOsed groups in figures 2, 4 and 5? The authors need to compare between these two groups.

Authors’ response: We now include supplement figures which illustrate the comparison of sedentary WT and Ostn-KO mice. We now also reference these figures in the results section and note that the findings here are similar to our previous findings that WT and Ostn-KO mice exhibit some differences in mitochondrial content and function at baseline which become much more prominent after exercise training. We explain these differences by the presence of a baseline level of physical activity in mice, even under regular housing conditions, such that there is a small degree of relevant musclin signaling in our ‘sedentary’ model.

2) Does synthetic musclin infusion has any effect on respiratory chain enzymes and respiratory capacity, mitochondrial biogenesis and cGMP/PKG signaling? The authors need to verify the molecular mechanism in vivo.

Authors’ response: The reviewer raises an interesting line of inquiry. We now include supplementary data and figures indicating that musclin infusion results in increased expression of mitochondrial DNA, as well as mRNA levels of CREB, PGC1α and ERRa which are important regulators of mitochondrial biogenesis. More detailed examination of synthetic musclin mechanisms for cardiac conditioning are planned for future focused pharmacologic studies, but were felt to be outside the main focus of this work that used synthetic musclin only as a tool to confirm the physiologic role of musclin and to rescue the Ostn-KO. 

3) The phosphorylation of CREB is important for its function, does musclin has any effect on the phosphorylation of CREB?

Authors’ response: Here we checked nuclear expression of PKGI as a marker of musclin induced cGMP signaling and write: “cGMP is known to activate protein kinase G (PKG) [ PMID: 35479330], which can translocate into the nucleus (and phosphorylates cAMP response element  binding (CREB) in baby hamster kidney (BHK) fibroblasts[PMID: 11175347; 15769622; 31509010] promoting its nuclear localization and ability to regulate transcription of peroxisome proliferator-activated receptor γ coactivator 1α (PGC1α)[ PMID: 11557984].”

4) The authors mentioned that musclin competitively interferes with binding to NRPc, allowing increased NP/NRPa binding, however, the experimental evidence is totally missing.

Authors’ response: We now further clarify that the competitive interference of musclin for the clearance receptor has been previously published (PMID: 19244276; 29326144; 17951249) and thus such experiments were not repeated here.

5) The expression level of LaminB1 in Figures 4A and 4C is not consistent.

Authors’ response: Figures 4 A, C include data from  different gels/blots, thus each gel/blot’s laminB1 or TBP reference used for normalization is shown immediately below the protein expression of interest. This has now been clarified in the methods: “For representative individual gel/blots, the reference protein (laminB1 and TBP) is shown immediately below the proteins of interest”.

6) The format of the references needs to be adjusted.

Authors’ response:  We have now downloaded the specific EndNote format for the journal and updated the references.

Round 2

Reviewer 1 Report

The authors have addressed my concerns. I endorse the manuscript for publication.

Reviewer 2 Report

The authors have answered my questions and comments satisfactorily. As a result of these changes, the paper is much improved and, in my opinion, suited for publication.